# Influence of Gelatin Source and Bloom Number on Gelatin Methacryloyl Hydrogels Mechanical and Biological Properties for Muscle Regeneration

**DOI:** 10.3390/biom13050811

**Published:** 2023-05-10

**Authors:** Mohammad B. Aljaber, Fiona Verisqa, Zalike Keskin-Erdogan, Kapil D. Patel, David Y. S. Chau, Jonathan C. Knowles

**Affiliations:** 1Division of Biomaterials and Tissue Engineering, Eastman Dental Institute, University College London, Royal Free Hospital Campus, Rowland Hill Street, London NW3 2PF, UK; mohammad.aljaber.19@ucl.ac.uk (M.B.A.); fiona.verisqa.19@ucl.ac.uk (F.V.); zalike.keskin.17@alumni.ucl.ac.uk (Z.K.-E.); d.chau@ucl.ac.uk (D.Y.S.C.); 2Dental Health Department, College of Applied Medical Sciences, King Saud University, Riyadh 11451, Saudi Arabia; 3UCL Eastman-Korea Dental Medicine Innovation Centre, Dankook University, Cheonan 31116, Republic of Korea; kapil.patel@ucl.ac.uk; 4Department of Nanobiomedical Science & BK21 PLUS NBM Global Research Center for Regenerative Medicine Research Center, Dankook University, Cheonan 31116, Republic of Korea; 5School of Cellular and Molecular Medicine, University of Bristol, Bristol BS8 1TD, UK

**Keywords:** hydrogel, muscle regeneration, GelMA, tissue engineering, mechanical properties

## Abstract

Approximately half of an adult human’s body weight is made up of muscles. Thus, restoring the functionality and aesthetics of lost muscle tissue is critical. The body is usually able to repair minor muscle injuries. However, when volumetric muscle loss occurs due to tumour extraction, for instance, the body will form fibrous tissue instead. Gelatin methacryloyl (GelMA) hydrogels have been applied for drug delivery, tissue adhesive, and various tissue engineering applications due to their tuneable mechanical properties. Here, we have synthesised GelMA from different gelatin sources (i.e., porcine, bovine, and fish) with varying bloom numbers, which refers to the gel strength, and investigated for the influence of the source of gelatin and the bloom number on biological activities and mechanical properties. The results indicated that the source of the gelatin and variable bloom numbers have an impact on GelMA hydrogel properties. Furthermore, our findings established that the bovine-derived gelatin methacryloyl (B-GelMA) has better mechanical properties than the other varieties composed of porcine and fish with 60 kPa, 40 kPa, and 10 kPa in bovine, porcine, and fish, respectively. Additionally, it showed a noticeably greater swelling ratio (SR) ~1100% and a reduced rate of degradation, improving the stability of hydrogels and giving cells adequate time to divide and proliferate to compensate for muscle loss. Furthermore, the bloom number of gelatin was also proven to influence the mechanical properties of GelMA. Interestingly, although GelMA made of fish had the lowest mechanical strength and gel stability, it demonstrated excellent biological properties. Overall, the results emphasise the importance of gelatin source and bloom number, allowing GelMA hydrogels to have a wide range of mechanical and excellent biological properties and making them suitable for various muscle tissue regeneration applications.

## 1. Introduction

Natural tissues are made of extracellular matrix (ECM) proteins in scaffold form, which supports cell anchoring and tissue assembly via integrin and other binding sites; collagen is the abounded protein in the skeletal muscle [1]. Muscles form approximately 45% of the mass of an adult human body and play a vital role in multiple dynamic events such as locomotion, mastication, ocular movement, as well as regulation of body metabolism [2]. Therefore, it is essential to restore muscle loss which can happen due to many reasons including myopathy, tumour extraction, and other accidental muscle injuries. One of the most common ways to restore volumetric muscle loss is by reconstructive surgeries, such as a muscle flap, which require an expert surgeon and are influenced by low transplant survival and donor site morbidity. A limited level of functionality can be restored after physiotherapy is done post-operation [3]. Tissue engineering (TEng) is another promising approach in which functional constructs made of biomaterials are assembled and biomolecules, cells, and drug molecules are utilised to repair, restore, and regenerate damaged/injured muscle tissues. TEng was defined by Langer and Vacanti [4] as “an interdisciplinary field that applies the principles of engineering and the life sciences toward the development of biological substitutes that restore, maintain, or improve tissue function”. The primary purpose of a scaffold is to mimic the ECM and allow cells to attach, migrate, and proliferate. Hydrogels are commonly used as a scaffold as they are hydrophilic polymers with a three-dimensional (3D) structure and viscoelastic properties mimicking those of biological tissues [5]. Hydrogels can absorb high amounts of water and allow a certain level of diffusion of oxygen and nutrients [6]. They can nicely mimic the native 3D structure of ECM by providing adhesion sites and signalling cues, which guide the formation of the desired tissue. Hydrogels can be natural, synthetic, or semi-natural, i.e., a chemically modified natural polymer such as gelatin methacryloyl (GelMA).

GelMA is a semi-natural hydrogel that results from the chemical reaction of gelatin and methacrylic anhydride (MA). The reaction was first introduced by Van Den Bulcke Al [7], who generated a product that combines the properties of natural and synthetic components. Primarily, it establishes excellent biological properties owing to the presence of natural cell binding motifs like arginine–glycine–aspartic acid (RGD), which play an essential role in cell attachment, growth, function, and differentiation [8]. Moreover, it also allows tuneable mechanical properties and can be conveniently photo-crosslinked by UV light. The amount of lysin groups replaced by MA during the reaction is usually referred to as the degree of substitution (DS). Controlling the DS is very important as it might influence the mechanical and biological properties of GelMA. Nichol and colleagues modified GelMA synthesis (porcine type A, 300 g) to improve its degree of substitution by introducing sequential addition of MA at a rate of 0.5 mL/min and reported a degree of substitution of around 53% [9]. Lee et al. further modified the synthesis to obtain better DS by changing the buffer to sodium carbonate-bicarbonate (CB) (pH 9.6) while adjusting pH to 9 before each time MA was added. One possible reason for the inefficient chemical synthesis of gelatin-MA is that a reaction by-product, methacrylic acid, lowers the solution pH during the reaction. In turn, gelatin’s free amino groups become ionised, inhibiting the reaction with MA. As a result, a DS of 97% was reported [10]. Shirahama and co-workers introduced the facile one-pot method by adding MA to dissolved gelatin (porcine type A, 175 g) all at once and adjusting the pH level to 9 right before adding MA and reported a DS of 96% [11].

Other factors have been shown to influence the mechanical properties of GelMA. Studies on the impact of GelMA concentrations suggested that an increase in the concentration increased the mechanical strength [9,11,12,13,14,15]. Other approaches to improve the mechanical properties of GelMA were evaluated, including conjugation of GelMA with other hydrogels such as PEG [16,17], alginate [18,19], pHEMA [20], or conductive hydrogel poly(aniline) [21]. The use of nanomaterials to improve the mechanical properties and electrical conductivity of GelMA was also reported by many researchers: for instance, graphene oxide [22], carbon nanotubes [23], and gold nano-particles [24]. Furthermore, GelMA hydrogel’s printability and processability have been evaluated in various reports [25,26,27,28].

GelMA is attractive in various tissue engineering applications due to its tuneable mechanical properties. Many researchers reported the use of GelMA in bone [12,29,30], nerve [31], cartilage [32,33], and corneal [34,35] tissue regeneration engineering. Regarding muscle regeneration, Hosseini and coworkers studied the influence of surface patterning and electrical stimulation on the differentiation and the alignment of myotubes using homemade GelMA that was synthesised from porcine type A gelatin with an unreported bloom number. It was observed that scaffolds with a surface pattern of 100 µm and electrical stimulation had improved myotube differentiation and alignment compared to 50 µm [36]. Costantini and fellow researchers studied the influence of GelMA (porcine type A, 300 g) stiffness on myogenesis. They reported a significant decrease in myotube formation as gel stiffness increased in 3D cell culture, in contrast to 2D cell culture. It was explained that 3D cell culture cells must digest the surrounding matrix to fuse with other cells to form a myotube, which was difficult in a dense structure [13].

Another factor that might influence the properties of GelMA is the type of gelatin used in the reaction. Gelatin can be obtained from animals such as bovine, porcine, and fish. Each type has different characteristics and different protein compositions. Gelatin can be extracted through acidic treatment (type A) or alkaline treatment (type B). The influence of the gelatin source on the rheological properties of low-concentration GelMA (~5% wt%) for bone regeneration purposes was reported [37]. They observed higher storage moduli in porcine than the others, with nearly 7 kPa, 3 kPa, and 0.4 kPa in porcine, bovine, and fish, respectively. We want to expand on that report and investigate the influence of gelatin source and bloom number on GelMA mechanical and biological properties for muscle repair and regeneration.

Previous research frequently fails to answer the question of how gelatin source types (bovine, porcine, and fish) and bloom number influence the mechanical and biological properties of GelMA, which will be addressed in this paper. Porcine gelatin type A has an isoelectric point (IEP) around pH 7–9, and was obtained through acid treatment at pH 1–2, while bovine gelatin type B was obtained through base treatment at pH 12–13 and has IEP at pH 5–6. This indicates different IEP points for each type. Thus, the degree of substitution of GelMA might vary depending on the gelatin type used, which might affect other properties such as biodegradation, swelling ratio (SR), porosity, and mechanical strength. Such differences have not yet been reported. In addition, gelatin comes with different strengths, known as bloom number. In 1925, Oscar T. Bloom invented the Bloom Gelometer, an apparatus devised for the purpose of measuring the stiffness of gelatin or gelatin films, thereby establishing a novel metric known as “bloom strength” [38]. Extracted from animal skin or bones through acid or alkaline treatment, collagen is released after a few days and then mixed with hot water to trigger denaturation into soluble gelatin. It is primarily the temperature of the water employed that controls the bloom strength, with a lower temperature resulting in higher bloom strength, which usually ranges from 50 g to 300 g [39]. However, the literature did not discuss a comparison between different bloom numbers of the same type of gelatin (e.g., porcine). Therefore, this work will also aim to study how bloom numbers of 175 g and 300 g porcine type A gelatin affect the mechanical strength of GelMA.

To address this concern, the GelMA-based hydrogel samples were characterised using proton nuclear magnetic resonance (^1^H-NMR), attenuated total reflectance Fourier-transform infrared spectroscopy (ATR-FTIR), scanning electron microscopy (SEM), dynamic mechanical analysis (DMA), swelling ratio (SR), and degradation tests, in addition to live/dead cell viability, metabolic activity, and cytotoxicity assessments.

## 2. Materials and Methods

### 2.1. GelMA Synthesis

Four different types of GelMA were prepared, which are 175 g bloom porcine GelMA (P-GelMA (175)), 300 g bloom porcine GelMA (P-GelMA (300)), bovine GelMA (B-GelMA), and fish GelMA (F-GelMA), while commercial GelMA made from porcine with 300 g bloom (C-GelMA) was used as a control. To synthesise GelMA, the following protocol was used, which was adapted from previous work [40]: 0.1 M of carbonate-bicarbonate (CB) buffer (Sigma-Aldrich, Merck Life Sciences UK Ltd., Poole, UK) (pH 9.6) was prepared by dissolving four capsules of CB in 200 mL of DI water. Then, 20 g of the appropriate gelatin (bovine, porcine (175), porcine (300), or fish) (Sigma-Aldrich, Merck Life Sciences UK Ltd., Poole, Dorset, UK) was added slowly to the CB buffer until completely dissolved. Methacrylic anhydride (MA) (Sigma-Aldrich, Merck Life Sciences UK Ltd., Poole, UK) was added at a rate of 200 µL/minute while adjusting the pH to 9.4 before each addition using 6 M of NaOH. The solution reacted for 3 h at a temperature of 55 °C under constant stirring at a speed of approximately 700 rpm with a Stuart US152 (Cole-Parmer, Stone, UK) stirrer. Then, the reaction was stopped by adjusting the pH to 7.4 using 6 M of HCL. To clear unreacted MA acid, the solution was dialysed using dialysis tubes with a molecular weight cut-off of 12,400 Da (Sigma-Aldrich, Merck Life Sciences UK Ltd., Poole, UK) for seven days against deionised water at 45 °C under constant stirring, while DI water was changed twice a day. After dialysis, the solution was placed in airtight plastic pots, sealed with parafilm, labelled, covered with aluminium foil, and stored at −80 °C until lyophilisation. Lyophilisation was performed for five days using Alpha 1–2 LDplus Martin Christ (SciQuip Ltd., Newtown Wem Shropshire, UK) until GelMA was completely dry, showing white solid material, which was the GelMA macromer. This was placed in plastic containers, sealed with parafilm to prevent rehydration, and stored at −80 °C in a dark environment covered with aluminium foil until use.

### 2.2. Sample Preparation

Freeze-dried GelMA macromer was dissolved in warm PBS (pH7.4) (50 °C) at (10% *w*/*v*) under continuous stirring. Once GelMA was dissolved entirely, (0.1% *w*/*v*) a lithium phenyl-2-4-6-trimethyl-benzoyl phosphinate (LAP) (Millipore Sigma, Merck Life Sciences UK Ltd., Poole, UK) photoinitiator was added under continuous stirring for about 15 min. To ensure uniform sample dimensions, 100 µL of the solution was pipetted into a 96-well plate, let cool to form physical crosslinking, and then cured inside a UV curing chamber (UV curing chamber, XYZ Printing, New Taipei City, Taiwan) using UV LED diodes with 405 nm at maximum intensity (level 3) for 2 min. Hydrogels were then removed from the plate for testing.

### 2.3. Mechanical Characterisation

#### 2.3.1. Proton Nuclear Magnetic Resonance

Proton nuclear magnetic resonance (^1^H-NMR) was performed to confirm the synthesis of GelMA by comparing the gelatin spectrum to GelMA to calculate the degree of GelMA substitution, which indicates how many amino groups were replaced by MA. An amount of 20 mg of each type of gelatin and GelMA were separately dissolved in 1 mL of deuterium oxide (D_2_O) (Sigma-Aldrich, Merck Life Sciences UK Ltd., Poole, UK). Spectra were collected using a Bruker Avance Neo 700 MHz (Bruker Ltd., Coventry, UK) at room temperature. All spectra were phase and baseline corrected before integrating the signals of interest (TopSpin software, version 3.5.7., Bruker Ltd., Coventry, UK), and the chemical shift scale was modified to account for the residual solvent signal (D_2_O (^1^H) = 4.79) [41]. The degree of substitution (*DS*) was calculated using the following Equation [32]:(1)$$DS\;\left(\%\right)=[1-\frac{\mathrm{lysine}\;\mathrm{methylene}\;\mathrm{proton}\;\mathrm{of}\;\mathrm{GelMA}}{\mathrm{lysine}\;\mathrm{methylene}\;\mathrm{proton}\;\mathrm{of}\;\mathrm{Gelatin}}]\times 100$$

#### 2.3.2. Fourier-Transform Infrared Spectroscopy

ATR-FTIR was done to ensure the successful synthesis of GelMA in comparison with the commercially available GelMA by identifying the presence of the specific vibrations of GelMA. It was used to assess the chemical composition of GelMA groups (P-GelMA (175), P-GelMA (300), B-GelMA, and F-GelMA) in comparison with C-GelMA. A photo-crosslinked sample of each hydrogel group was placed on the diamond of an attenuated total reflectance (ATR) (Golden Gate ATR, Specac Ltd., Orpington, UK) machine and characterised using a PerkinElmer Spectrum One spectrometer (PerkinElmer Ltd., Seer Green, Beaconsfield, UK) in the 4000–600 cm^−1^ wavenumber region. The data presented in spectra show peaks of specific chemical groups found in the gel samples.

#### 2.3.3. Scanning Electron Microscopy

To characterise the internal microstructures of the GelMA hydrogels, the crosslinked hydrogel samples were frozen at −80 °C and then lyophilised. The dried samples were frozen at −20 °C and cross-sectioned, then coated with 95% gold and 5% palladium by Polaron E5000 Sputter Coater (Quorum Technologies, Lewes, UK), and images were taken by using a Philips XL30 field emission SEM (Amsterdam, The Netherlands). ImageJ software was used to measure pore size based on scanning electron microscopy (SEM) images.

#### 2.3.4. Dynamic Mechanical Analysis

Compressive mechanical testing was performed on GelMA samples using a Discovery DMA 850 (TA Instruments, Elstree, Hertfordshire, UK) to study how GelMA hydrogel strength is affected by gelatin bloom number and gelatin source. GelMA samples were prepared according to the previously described protocol. After that, they were immersed in PBS (pH 7.4) at 37 °C for 24 h to reach equilibrium. Excess water was carefully dabbed off the sample surface. Then, compressive testing was performed at a crosshead speed of 0.5 mm/min at 37 °C. In addition, the DMA was used to measure the oscillation time sweep at 37 °C for 5 min using a 1 Hz frequency and 0.1% strain in order to investigate the viscoelastic properties of the crosslinked hydrogels. Sample dimensions were 6 mm in diameter and 3 mm in thickness. Stress and strain curves were obtained, and compressive strength and Young’s modulus were calculated.
(2)$$Young^{\prime }s\;modulus\;\left(elastic\right)=\frac{stress}{strain}$$

#### 2.3.5. Gelation Temperature

The rheology temperature sweep test was carried out using a HAAKE™ Viscotester™ iQ Rheometer (Fisher Scientific UK Ltd., Loughborough, UK) in order to study the gelation temperature of different gelatin types as well as to study the influence of methacrylation on gelling temperature. An amount of 1 mL of hydrogel was placed on the base and then the test was done using 40 °C decreased to 4 °C at 1% strain and 1 Hz frequency at a rate of 1 °C/min.

#### 2.3.6. Swelling Ratio

*SR* was carried out to calculate the liquid uptake of the hydrogels. It was performed to investigate the influence of gelatin bloom number (175 g and 300 g porcine type A) and gelatin source (porcine, bovine, and fish) on the swelling properties of GelMA. Hydrogel samples were crosslinked, then freeze-dried to measure their dry weight, and then immersed in PBS (pH 7.4) for 24 h at 37 °C to measure their wet weight (n = 5). It is important to mention that before measuring the weights of samples, the excess surface water was carefully removed by blue tissue (Kimberly-Clark™ WypAll™ Fabric Wipers, Fisher Scientific UK Ltd., Loughborough, UK). The SR of a sample was calculated as follows:(3)$$SR=\frac{W1-\;W2}{W2}$$
where W2 is the initial dry weight and W1 is the wet weight of hydrogel samples [42].

#### 2.3.7. Degradation Test

GelMA hydrogels are biodegradable materials. In this study, the rate of GelMA hydrogel degradation was evaluated. Hydrogel samples were prepared from P-GelMA (175), P-GelMA (300), B-GelMA, F-GelMA, and C-GelMA, then crosslinked by UV for 2 min (n = 5). Samples were incubated in PBS at 37 °C for 24 h to reach equilibrium. The initial sample weight was recorded (W1). The weight of the samples was recorded every day at the same time point for up to 2 weeks (W2). The remaining weight was calculated according to [25]:(4)$$\mathrm{Remaining}\;\mathrm{weight}\;(\%)=[100+\frac{\left(W2-W1\right)}{W1}]\times 100$$

### 2.4. Biological Characterisation

#### 2.4.1. C2C12 Encapsulation

C2C12 muscle cell line cells were cultured in Gibco Dulbecco’s modified Eagle’s medium (DMEM) high glucose (Fisher Scientific UK Ltd., Loughborough, UK) supplemented with 10% FBS, 2 mM L-Glutamine, 100 U/mL penicillin, and 100 μg/mL streptomycin (PS) (Sigma-Aldrich, Merck Life Sciences UK Ltd., Poole, UK). GelMA hydrogel solution was mixed with LAP until wholly dissolved and sterilised by being passed through a 0.22 μM syringe filter (Fisher Scientific UK Ltd., Loughborough, UK). Next, 100 µL of GelMA was added to each well in a 96-well plate. An amount of 2.5 × 10^4^ of C2C12 cells were added to each sample, followed by gentle mixing to avoid air bubbles. Then, the well plate was placed in a UV curing chamber for 2 min for crosslinking before being incubated at 37 °C in the presence of 5% CO_2_.

#### 2.4.2. AlamarBlue™

AlamarBlue™ (AB) Cell Viability Assay (Invitrogen™, Fisher Scientific UK Ltd., Loughborough, UK) was used to assess C2C12 cell viability. AB is a Resazurin-based assay, which is a non-toxic, blue-coloured reagent. Resazurin is converted to resorufin, a red-coloured and highly fluorescent molecule, when it enters a living cell. The protocol was utilised according to the manufacturer’s instructions. Briefly, 10 µL of AB reagent was added into each well at a ratio of (1:10) of the growth medium. After 4 h of incubation at 37 °C, the fluorescence intensity was measured with a Biotek FLx800 microplate reader (Biotek UK, Swindon, UK) at an excitation wavelength of 560 nm and emission wavelength of 590 nm.

#### 2.4.3. Lactate Dehydrogenase

Lactate dehydrogenase (LDH) is an enzyme released by cells following a compromised/damaged membrane. The CytoTox-ONE Homogeneous Membrane Integrity assay kit (Promega, Southampton, UK) was used according to the manufacturer’s instructions to measure the released LDH. Briefly, 50 µL of CytoTox-ONE was added to each well in a 96-well plate containing 50 µL of the supernatant at a ratio of (1:1). The well plate was incubated at 22 °C for 10 min. Then, 50 µL of Stop solution was added to each well at a ratio of (1:2). The 96-well plate was read at an excitation wavelength of 560 nm and emission wavelength of 590 nm using a Biotek FLx800 microplate reader. It was important to include a background (negative) control, which was triplicate wells with growth medium only (GM). Furthermore, it was important to have a positive control, which was triplicate wells containing 2 μL of Triton X-100, to kill cells and obtain maximum LDH release. To calculate the percentage of cytotoxicity, the following equation was used:(5)$$\%\;of\;cytotoxicity=[\frac{\left(Experimental-GM\;background\right)}{\left(Maximum\;LDH\;release-GM\;background\right)}]\times 100$$

#### 2.4.4. Live/Dead

A LIVE/DEAD™ imaging kit (488/570) (Fisher Scientific UK Ltd., Loughborough, UK) was used to visualise live and dead cells on the surface of the samples. The kit comprises two components: calcein-AM, which is membrane-permeable, binds with the cytoplasm, and fluoresces green in living cells; and ethidium homodimer-1, which is membrane-impermeable and binds with DNA in dead cells and fluoresces red. The protocol was utilised according to the manufacturer’s instructions. Briefly, the old medium was removed, then samples were washed with PBS. Fresh medium with 10% live/dead reagent was added to samples in a dark environment. The 96-well plate was incubated for 30 min at 20–25 °C. After incubation, cells were imaged using an inverted fluorescence microscope (Leica DM IRB, Milton Keynes, UK) to determine live and dead cells.

### 2.5. Statistical Analysis

The results were statistically analysed using one-way analysis of variance (ANOVA) with Tukey’s post hoc test (GraphPad Software, version 9, San Diego, CA, USA); *p* < 0.05 was considered to be statistically significant.

## 3. Results

### 3.1. GelMA Synthesis

Several batches of GelMA were synthesised, and the hydrogels were prepared according to the previously mentioned protocols. Then, GelMA cylinder-shaped samples were prepared with 6 mm diameter and 3 mm thicknesses. All hydrogel groups showed the same clear hydrogel sample after UV crosslinking (Figure 1).

### 3.2. Proton Nuclear Magnetic Resonance

^1^H-NMR was used to confirm the presence of methacryloyl groups as well as to define and calculate the degree of substitution. An amount of 20 mg of each gelatin and GelMA were dissolved separately in 1 mL of D_2_O (Figure 2) and were analysed. The spectrum of GelMA shows new peaks compared to gelatin: the peak at (3 ppm) indicates the non-modified lysine of gelatin, the peak at (1.8 ppm) represents acrylic proton methacrylamide, and the two peaks at (≈5.5 ppm) are an indication of the presence of vinyl protons of MA. The degree of substitution was calculated to be 87, 83, 63, and 95% for P-GelMA (175), P-GelMA (300), B-GelMA, and F-GelMA, respectively.

### 3.3. Attenuated Total Reflectance Fourier-Transform Infrared Spectroscopy

The ATR-FTIR spectra of the homemade GelMA groups (B-GelMA, F-GelMA, P-GelMA (300), and P-GelMA (175)) display similar peaks as in the commercially available (C-GelMA). The spectra (Figure 3) indicate the presence of the following functional groups: a broad peak at ~3302 cm^−1^ is the standard signal for (N-H) and (O-H) stretching, and a peak at ~1644 cm^−1^ for amine (N-H).

### 3.4. Scanning Electron Microscopy

GelMA samples were crosslinked by UV for 2 min, then freeze-dried. The dried samples were frozen at −20 °C before cutting them in half using a blade. An SEM microscope was used to visualise the cross-section to look at the inner structure of the hydrogels. The inner structure (Figure 4) displayed a porous structure with an average pore size of 28.5 ± 8.3 µm, 33 ± 10.2 µm, 36.5 ± 10.3 µm, 29.2 ± 26.2 µm, and 32.2 ± 4.5 µm for P-GelMA (175), P-GelMA (300), B-GelMA, F-GelMA, and C-GelMA, respectively.

### 3.5. Dynamic Mechanical Analysis

Hydrogel samples were prepared cylinder-shaped (6 mm, 3 mm) and then incubated in PBS for 24 h to reach equilibrium. An oscillation time sweep test was performed for 5 min at 37 °C using 1 Hz frequency and 0.1% strain. The results indicate that B-GelMA had significantly higher storage and loss moduli with approximately 25 KPa and 3 KPa, respectively (Figure 5B). In addition, a compressive test was also carried out at 37 °C, which also proved that B-GelMA was the highest among the groups, and F-GelMA was the lowest, with approximately 60 and 10 kPa, respectively (Figure 5A).

### 3.6. Gelation Temperature

After GelMA was prepared at 10% (*w*/*v*), 1 mL was needed to test the rheological properties of the gels. The temperature sweep test displayed an increase in the storage and loss modulus values of the gels as the temperature decreased from 40 °C to 4 °C. This change in value represents a change in the structure of the gel, which indicates that the gel is starting to physically crosslink due to being thermo-responsive hydrogel. A 175 g porcine gelatin seemed to start forming gel at 23 °C (Figure 6A,C,E,G), 300 g porcine gelatin at 27 °C, and bovine gelatin at 25 °C, while fish gelatin did not form a gel at all. The results also emphasise the influence of the reaction between MA and gelatin to decrease GelMA gelling temperature of P-GelMA (175) from ≈23 °C to 20 °C, P-GelMA (300) to ≈ 19 °C, and B-GelMA to ≈14 °C (Figure 6B,D,F).

### 3.7. Swelling Ratio

A one-way ANOVA was performed to study the influence of the gelatin source and bloom number on the SR of the hydrogels. The results (Table 1) revealed that there was a statistically significant influence on the SR among the groups (F (4, 10) = 61.08, *p* < 0.001). Tukey’s HSD Test for multiple comparisons found that the mean value of SR was significantly different between P-GelMA (175) and P-GelMA (300) (*p* < 0.001), emphasising the influence of the gelatin bloom number. Furthermore, Tukey’s HSD Test for multiple comparisons found that the mean value of SR was significantly different between B-GelMA, F-GelMA, and C-GelMA (*p* < 0.0001), emphasising the influence of the gelatin source.

### 3.8. Degradation Test

Degradation test results of GelMA samples incubated in PBS at 37 °C reveal that B-GelMA had a significantly slower degradation rate. In contrast, F-GelMA had the fastest degradation rate, completely degraded by day 9 (Figure 7).

### 3.9. Metabolic Activity and Cytotoxicity

AlamarBlue was used to assess the metabolic activity of C2C12 cells cultured within GelMA samples and LDH to assess the cytotoxicity. In these studies, the control group was C2C12 cells cultured on tissue culture plastic. Metabolic activity data (Figure 8A) for 3D cell culture were low in all groups but increased significantly on day 4 and continued to increase up to day 14. Low LDH release was observed on days 1 and 4 (<10%) and slightly increased by day 7 to ~10% due to reaching full confluency (Figure 8B).

### 3.10. Live/Dead Images

The cell viability of C2C12 cells encapsulated within GelMA hydrogels was explored by an L/D imaging kit. L/D images were taken using the fluorescence microscope on days 1, 3, and 7. As shown in (Figure 9), cell viability increased over the culture in all groups.

## 4. Discussion

GelMA hydrogels have gained significant interest in the past few years for muscle regeneration. They have outstanding biological properties and tuneable mechanical properties. GelMA comes in many types, each made from different gelatin sources. GelMA can be made from porcine, bovine, and fish gelatin. Each type can have different characteristics, such as different colours, viscosities, gelling temperatures, and amounts of amino acids. Such differences might have an influence on GelMA’s biological and mechanical properties, which will be discussed.

GelMA hydrogels from different sources and bloom numbers were successfully synthesised (Figure 1), as confirmed by the ^1^H-NMR results. The results indicated (Figure 2) that GelMA hydrogels prepared using 300 g porcine and 175 g gelatin demonstrated a similar DS of 87% and 83%, respectively. These findings suggest that the bloom number did not significantly influence the DS of GelMA hydrogels, while the DS of GelMA made from different gelatin sources showed significant differences among the groups, with DS values of 83%, 63%, and 95% with a porcine, bovine, and fish origin, respectively. These differences are due to each gelatin type having varying IEP values, which is a result of each kind being prepared through different processes. Acidic treatment in the case of porcine gelatin and alkaline treatment in the case of bovine gelatin resulted in IEP values of 7–8 and 5–6 for porcine and bovine gelatin, respectively [43,44]. These findings suggest that the gelatin source has a significant influence on the DS of GelMA. The reaction pH was kept above nine during the addition of MA according to previous work [40], while increasing pH to higher levels may lead to faster methacrylic group degradation [45]. Doing the reaction at another pH value might have a different influence on the DS of GelMA. The degree of GelMA substitution is crucial as it may influence the other characteristics of GelMA. It has been shown in the past that a higher DS of GelMA can lead to a greater compressive modulus and a smaller pore size [9]. However, high DS was found to be less favourable for cell viability when compared to low DS.

Furthermore, FTIR was done to evaluate the synthesis of homemade GelMA (B-GelMA, F-GelMA, P-GelMA (300), and P-GelMA (175)) by comparing the spectra with the commercially available GelMA, which is made of 300 g bloom porcine gelatin. The results in (Figure 3) show that all groups were successfully synthesised, as they presented similar peaks to the commercial GelMA, illustrating the presence of gelatin and MA groups. The GelMA spectrum obtained here is similar to that shown previously in the literature [22,42,46,47].

The internal structure of GelMA, which was investigated by SEM (Figure 4), presents a uniform porous structure in all GelMA types except for F-GelMA. F-GelMA illustrated a wide range of pores with 29.2 µm ± 26.2 µm. This perhaps is a result of F-GelMA not forming physical crosslinking by temperature, unlike the other groups, which was proven by the rheological analysis (Figure 6H). The physical crosslinking, when applied prior to UV crosslinking, could significantly enhance the mechanical integrity of GelMA and, consequently, the internal structure, as shown in previous work [35]. In addition, F-GelMA demonstrated a very high degree of substitution (Figure 2), which was proven by [48,49,50] also to influence the porous structure of GelMA hydrogels.

In order to characterise the mechanical properties of these hydrogels to emphasise the influence of gelatin source and bloom number, several studies were carried out, such as DMA, rheology, swelling ratio, and degradation tests. DMA was used to measure the compressive strength and the storage and loss moduli of GelMA samples at 37 °C. The compressive strength of the homemade GelMA, specifically in P-GelMA (300) and B-GelMA, presented significantly higher values than the commercially available one (C-GelMA). The compressive strength is essential for gel stability during and after scaffold implantation. Furthermore, the bloom number, referred to as the gel strength, was observed to influence the compressive strength of the final product of GelMA. The results indicated that P-GelMA (300) had higher compressive strength compared to P-GelMA (175), with values of 40 and 18 kPa (Figure 5A), respectively. In other words, using a higher bloom number of gelatin will result in higher GelMA compressive strength. Interestingly, the source from which the gelatin was obtained was also found to affect the GelMA strength. B-GelMA, usually type B, obtained through alkaline treatment, had the highest compressive strength (60 kPa) compared to the other groups. This illustrates the importance of the gelatin source, as the compressive strength could range from ~5 kPa in F-GelMA to 60 kPa in B-GelMA. This wide range of strength makes GelMA attractive in many TEng applications. These findings were similar to what had been presented in the past in the case of GelMA made of porcine gelatin [51] and bovine gelatin [42]. The viscoelastic properties of the hydrogels were investigated by measuring the storage (elastic) modulus and the loss (viscous) modulus. The storage modulus increased (Figure 5B) as the bloom number of gelatin increased (8 kPa and 12 kPa for P-GelMA (175) and P-GelMA (300), respectively). B-GelMA hydrogels showed the highest elastic properties among the groups with 25 kPa. This indicates that GelMA made of bovine gelatin has more deformation resistance than GelMA made of porcine and fish gelatin.

Gelation temperature was found to be different according to the type of gelatin. In the case of porcine gelatin, it was demonstrated that the higher the bloom number, the higher the gelation temperature (23 °C and 27 °C for 175 g and 300 g gelatin, respectively) (Figure 6A,C). The gelation temperature of bovine gelatin was 25 °C. In the case of fish gelatin, no gelation was observed up to 4 °C. The influence of MA and gelatin reaction on the gelling temperature was also investigated. It was observed that the addition of MA reduced the gelation temperature to 20 °C in P-GelMA (175), 19 °C in P-GelMA (300), and 14 °C in B-GelMA (Figure 6B,D,F). This is due to replacing the lysine groups in gelatin with MA. This decrease in gelling temperature is important as it allows GelMA to be used and processed at room temperature. Knowing the gelling temperature of each type is essential in improving the processability of these hydrogels, especially in the applications of 3D printing and bioinks. Furthermore, it also allows physical crosslinking, which was proven to enhance the mechanical properties of the hydrogels.

The results of the swelling ratio study (Table 1) indicated that P-GelMA (300) had a higher SR compared to P-GelMA (175), which means that the higher the bloom number, the higher the SR. Both groups had a similar degree of substitution values, which eliminates the influence of DS and emphasises the impact of bloom number. Other results comparing the source of gelatin revealed that B-GelMA had the highest SR compared to the other groups. This is due to B-GelMA having the lowest DS of 63% (Figure 2). The influence of DS on SR was demonstrated in the past: authors of [9,42,52] reported that the SR of GelMA hydrogels would decrease as the DS increases. The gelatin source might also influence the SR as porcine gelatin is usually obtained through acid-hydrolysis of collagen, while bovine type B gelatin is usually obtained through base-hydrolysis, which also affects the protein and fat component in each type and thus influences the other mechanical properties [53].

To further understand the influence of bloom number and gelatin source of GelMA, the biodegradability of these hydrogels was assessed. GelMA is known to be a biodegradable hydrogel. However, each type may have a different degradation rate. The results of the degradation tests, which were performed by incubating samples in PBS at 37 °C, showed that F-GelMA degraded by day 9 (Figure 7), which was faster than all the other types owing to the weak mechanical and physical properties of fish gelatin. Fish gelatin was the only type that did not demonstrate physical crosslinking by temperature. In contrast, other types would crosslink at room temperature. This emphasises the importance of physical crosslinking in enhancing the mechanical properties of GelMA. It has been reported that physical crosslinking of GelMA at 4 °C prior to UV crosslinking led to improved mechanical properties [35]. The results also indicated that B-GelMA had the slowest degradation rate among the groups; it had >70% remaining weight by day 13. This, perhaps due to B-GelMA’s superior mechanical properties, resulted in more stable hydrogel (Figure 5A). With that being said, it is important to mention that taking gels out to weigh them every day could have accelerated the gels’ degradation rates. In (Figure 8A), metabolic activity of GelMA hydrogels were presented up to day 14, which did not lead to hydrogel degradation.

In the case of 3D cell culture where cells were cultured within the hydrogel samples, the AB results indicate that all the homemade GelMA groups presented good metabolic activity with culture time compared to C-GelMA (Figure 8A). These findings were also confirmed by LDH assay. The assay indicated a deficient LDH release by day 7 (Figure 8B). These results prove how good the biological properties of GelMA hydrogels are in 3D cultures. Cell viability was also assessed by live/dead imaging to visualise the living and dead cells. The images illustrate the increase of cell viability in all GelMA groups by day 7 (Figure 9). These results parallel what has been found in the literature, illustrating the good biological properties of GelMA [13,17,54]. The biological characterisation concluded that gelatin source and bloom number did not significantly influence the cell viability of C2C12 cells encapsulated within the hydrogels. However, we believe that differentiation studies of C2C12 to investigate protein and gene expressions of myotube formation are important to evaluate in the future.

## 5. Conclusions

In conclusion, GelMA is influenced by different bloom numbers and the source of origin of gelatin. The B-GelMA was proven to have superior compressive mechanical properties compared to the other types of GelMA made of porcine and fish. A solid construct is crucial as it will aid during scaffold implantation and give cells support to attach and proliferate after the implantation. It also demonstrated a significantly higher SR and lower degradation rate, which enhance hydrogels’ stability, allowing enough time for cells to proliferate and differentiate to restore muscle loss. All GelMA types exhibited excellent biological properties in terms of high cell viability, metabolic activity, and low cytotoxicity. The bloom number has influenced the mechanical properties of GelMA as well. P-GelMA (300) had a significantly higher swelling ratio and compressive strength.

## Figures and Tables

**Figure 1 biomolecules-13-00811-f001:**
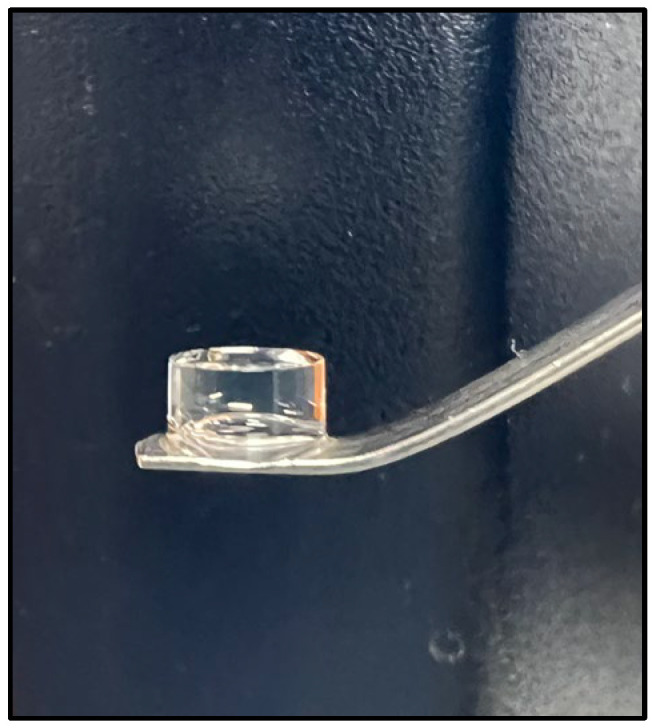
A picture of a photo-crosslinked GelMA hydrogel sample after it was removed from the 96-well plate.

**Figure 2 biomolecules-13-00811-f002:**
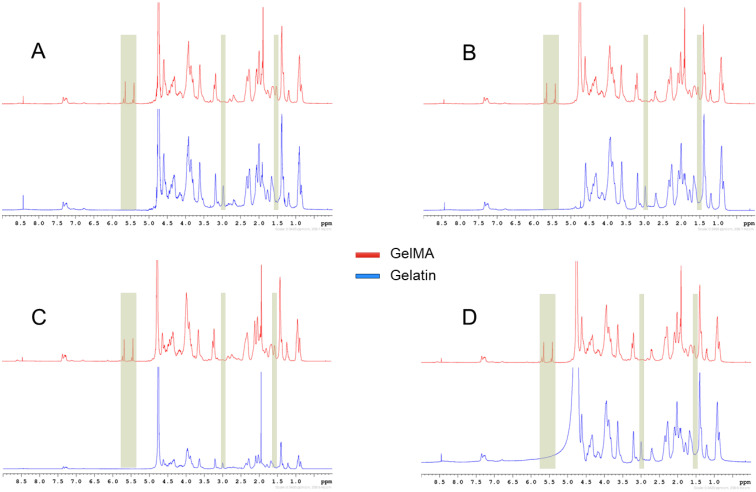
^1^H-NMR spectra of gelatin and GelMA illustrating the new peaks in GelMA, (**A**) P-GelMA (175), (**B**) P-GelMA (300), (**C**) F-GelMA, and (**D**) B-GelMA. The chemical shift at (3 ppm) represents the lysine group in gelatin which was replaced by MA in GelMA, while the other highlighted chemical shifts (≈5.5 and 1.6) show the new peaks in GelMA representing vinyl protons of MA and acrylic proton methacrylamide, respectively.

**Figure 3 biomolecules-13-00811-f003:**
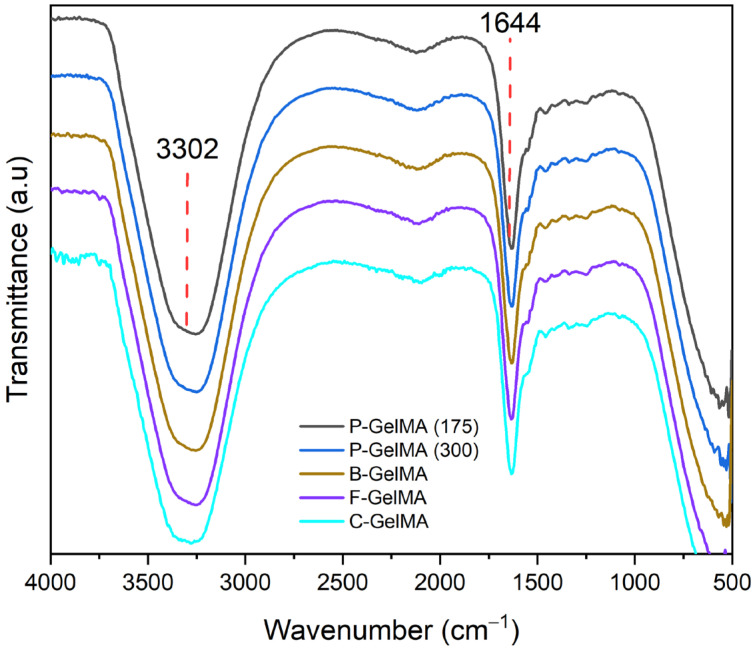
ATR-FTIR spectra of P-GelMA (175), P-GelMA (300), B-GelMA, and F-GelMA in comparison with C-GelMA.

**Figure 4 biomolecules-13-00811-f004:**
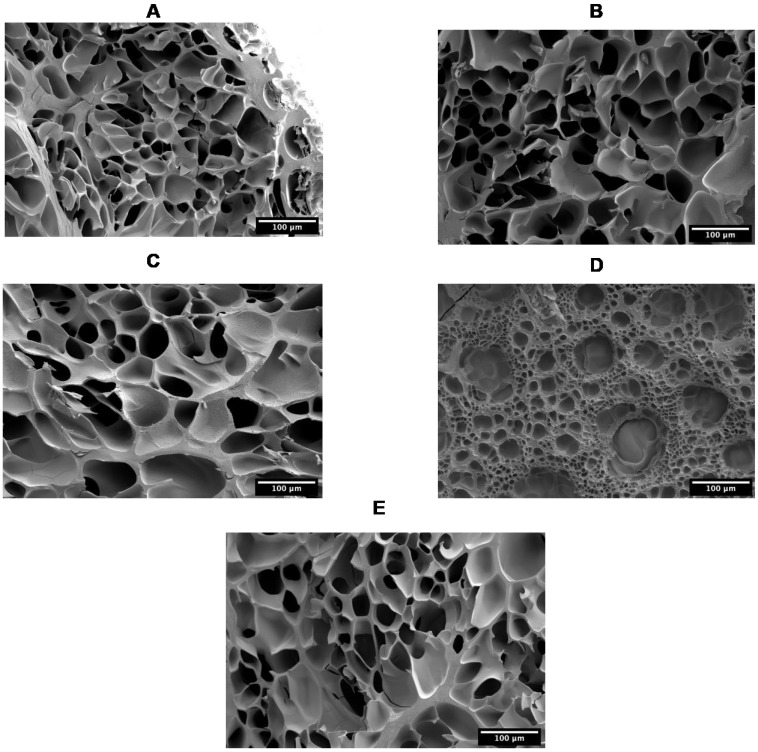
SEM images of the internal microstructure of (**A**) P-GelMA (175), (**B**) P-GelMA (300), (**C**) B-GelMA, (**D**) F-GelMA, (**E**) C-GelMA. Scale bar: 100 µm.

**Figure 5 biomolecules-13-00811-f005:**
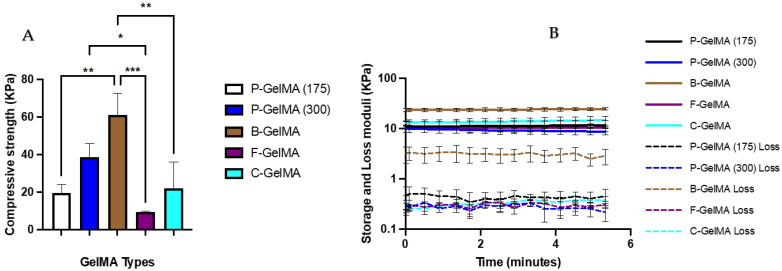
(**A**) Average DMA compressive strength of different GelMA groups immersed in PBS for 24 h at 37 °C to reach equilibrium (n = 5). The test was performed at 37 °C. (**B**) Average storage and loss moduli of GelMA samples using the same conditions after 5 min of time sweep test (n = 5), * *p*< 0.05, ** *p* < 0.01, *** *p* < 0.001.

**Figure 6 biomolecules-13-00811-f006:**
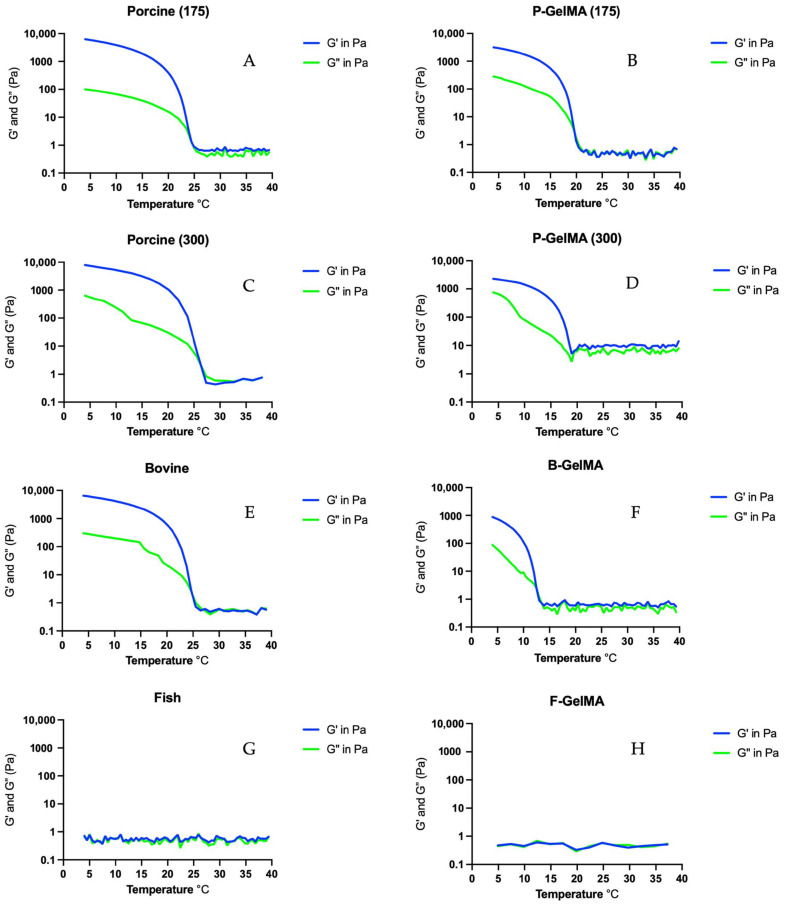
Temperature sweep results of (**A**) 175 g porcine gelatin, (**B**) P-GelMA (175), (**C**) 300 g porcine gelatin, (**D**) P-GelMA (300), (**E**) bovine gelatin, (**F**) B-GelMA, (**G**) fish gelatin, and (**H**) F-GelMA decreased from 40 °C to 4 °C.

**Figure 7 biomolecules-13-00811-f007:**
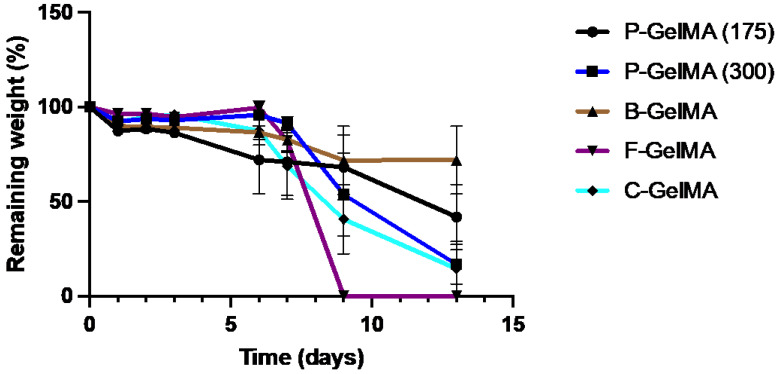
Remaining weight of different GelMA types after 13 days of incubations in PBS at 37 °C (n = 5).

**Figure 8 biomolecules-13-00811-f008:**
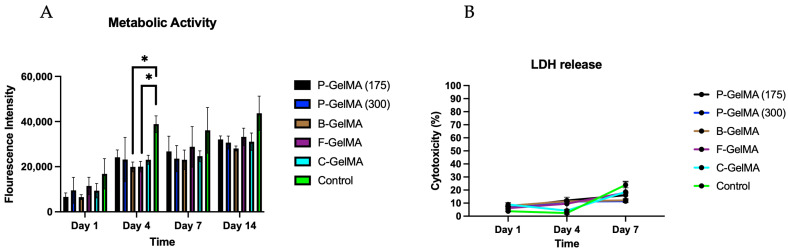
Cell viability and cytotoxicity of different GelMA groups (n = 5), (**A**) metabolic activity of 3D cell culture using AB, (**B**) LDH release of 3D cell culture. * *p* < 0.05.

**Figure 9 biomolecules-13-00811-f009:**
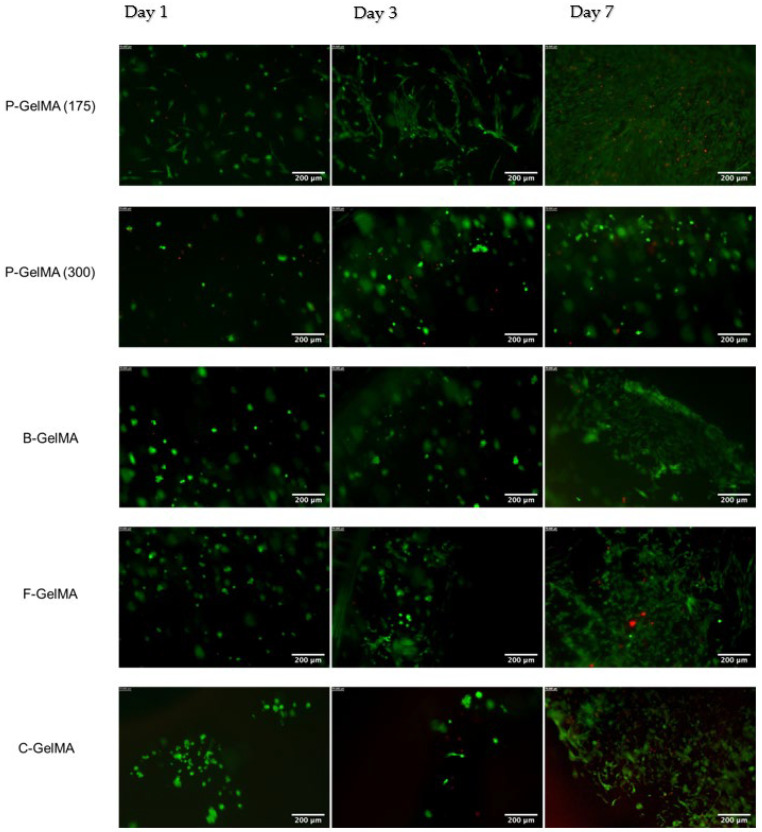
Live/Dead images of P-GelMA (175), P-GelMA (300), B-GelMA, F-GelMA, and C-GelMA on days 1, 3, and 7 using the fluorescence microscope, indicating living cells (green) and dead cells (red). Scale bar: 200 µm.

**Table 1 biomolecules-13-00811-t001:** The swelling ratio of different types of GelMA after 24 hrs of incubating dried samples in PBS at 37 °C (n = 5), showing significant differences among the study groups (*p* < 0.001) except between F-GelMA and C-GelMA, for which there was no significant difference (ns).

GelMA Type	Swelling Ratio (%)
P-GelMA (175)	523.92 ± 71.92
P-GelMA (300)	863.20 ± 30.75
B-GelMA	1098.87 ±64.63
F-GelMA	786.47 ± 24.67 *
C-GelMA	699.78 ± 11.10 *

^*^ no significance difference on comparison.

## Data Availability

Data will be made available upon request to the corresponding author.

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
