# Peer review of "Influence of Gelatin Source and Bloom Number on Gelatin Methacryloyl Hydrogels Mechanical and Biological Properties for Muscle Regeneration"

_biomolecules, 2023, doi:10.3390/biom13050811_

Round 1

Reviewer 2 Report

This interesting paper is a characterization of various types of GelMA hydrogels fabricated from different sources of gelatin. The investigation of the relationship between bloom strength and gel mechanical properties is interesting. The paper itself is well-written, polished, and has a significant amount of data relative to the rheological and mechanical properties of the gels. The results presented are a useful reference for labs working with GelMA hydrogels and the paper can almost be regarded as a methods paper, with broad applicability to labs that make their own materials. 

With that being said, my main concern with this paper is in the assessments of cytocompatibility and on the utility of these gels for muscle tissue engineering. The authors use C2C12 cells to confirm that the gels are cytocompatible, and grow the C2C12 cells in them for up to one week. In my opinion, this is not enough time to demonstrate the cytocompatibility needed for muscle tissue engineering. I realize that these gels degrade after about 9-13 days; however, this is something that should also be addressed, as a gel that falls apart in less than 2 weeks is also unlikely to be useful for most tissue engineering applications - perhaps it is more useful as a scaffold for a bioink. Additionally, in this reviewer's experience, C2C12 cells will survive almost anything. Making them live in a gel for a week is not indicative of particularly high cytocompatibility. Furthermore, no assessment of C2C12 differentiation, spontaneous or otherwise, was made in the paper. 

To be clear, these are issues that the authors may be planning to address in future directions and a future publication. I do not think all these experiments necessarily need to be added into this paper, but the discussion could do a better job of acknowledging some of these limitations. If the authors do have data on the C2C12 differentiation potential in this study, it would make a very nice addition to the existing data. 

Additionally, there are some grammatical errors/typos throughout that should be corrected before publication. 

Reviewer 3 Report

The scientific quality of the manuscript is insufficient for publication in its current form. 

1.      The introduction is too long, the authors should make it more concrete.

2.      It is still difficult to find the novelty of the work concerning what has already been published. What is the difference between what is published with what the authors want to publish? It is not clear, and the authors must describe these differences in the introduction section.

3.      Figure 1 looks like a meaningless drawing, and the authors should make a more scientific figure. For example, the authors can place the amino acids of each polypeptide chain and functional groups, and enzymes that involve in the process, etc.

4.      Line 226. Why were the samples again cross-linked by UV treatment?

5.      During the fabrication process, the hydrogels were obtained (2.2. Sample preparation). In line 234, the authors evaluate the gelation temperature of different gelatin types as well as study the influence of methacrylation on gelling temperature. I don't understand why they do this evaluation if the hydrogels have already been formed..... The authors want to determine the melting point of the hydrogels????' The crosslinking process is supposed to be an irreversible process..... Explain more in detail.

6.      Line 245. What pH value has PBS buffer?

7.      In the swelling ratio, the authors should report as a percentage. The authors can revise DOI: 10.1002/jbm.a.36794.

8.      The hydrogels must evaluate the viscoelastic properties of the hydrogels at 37 °C. The authors can revise DOI: 10.1007/s10853-023-08385-8

9.      Line 298. Why calcein-AM fluoresces green in living cells? What is the mechanism?

10.  The results and discussion sections are poor. More comparisons with previous literature should be discussed.

11.  The authors must add photographs of the hydrogels obtained in the results section.

12.  Line 329. In the ATR-FTIR results, the authors must explain how the hydrogels were produced. Do hydrogels were physically or chemically crosslinked? In this section, the authors must demonstrate it.

13.  Line 338. Why did the hydrogels show different pore sizes?

14.  The results in Figure 6 are inferior, and the authors should improve the quality of the rheological measurement. In the figure, a lot of noise is observed.

15.  Table 1 and Figure 8 must display if significant differences exist between simples using numbers or letters.

Minor editing of English language required

Round 2

Reviewer 3 Report

The manuscript can be accepted